# Octogenarian patients and laser-assisted lead extraction: Should we put a limit?

**Sameer Al-Maisary** *, **Gabriele Romano, Matthias Karck, Raffaele De Simone, Jamila Kremer, Rawa Arif**

Department of Cardiac Surgery, Heidelberg University Hospital, Heidelberg, Germany

* yemendoctor@yahoo.com

**Data Availability Statement:** All relevant data are within the paper and its Supporting information files.

**Funding:** The author(s) received no specific funding for this work.

## Abstract

### Background

Transvenous laser lead extraction (TLE) for cardiac implantable electric devices (CIED) is a challenging procedure especially if performed in octogenarians. In this study we evaluated the safety and efficacy of transvenous laser lead extraction in elderly patients.

### Methods

This is a retrospective study of octogenarian patients who underwent laser-assisted lead extraction (LLE) (GlideLight laser sheath, Philips, San Diego, USA). 270 Consecutive patients were included. Patients were divided into two groups. Octogenarian group and non-octogenarian group. The Data was gathered from patients treated between September 2013 and January 2020 and is retrospectively analyzed.

### Results

Of 270 consecutive patients, 38 (14.0%) were 80 years old or more. The total number of the extracted leads was 556 among which 84(15.0%) from the Octogenarian group. From these leads were 155 single coil leads, 82 dual coil leads, 129 right ventricular pacing leads, 155 right atrial leads, and 35 left ventricular leads. In the Octogenarian group the number of removed leads was as follows: 13 single coil leads, 10 dual coil leads, 28 right ventricular pacing leads, 28 right atrial leads and 5 left ventricular leads. No mortality was recorded in the Octogenarian group. One patient in the YG suffered from a superior vena cava tear and one patient suffered from pulmonary embolism.

### Conclusion

In octogenarian laser assisted lead extraction patients is a safe and effective procedure. No increase in morbidity, mortality or perioperative complication could be recorded in this group. Age should not be a limiting factor to perform this procedure.

**Competing interests:** The authors have declared that no competing interests exist.

## Introduction

Cardiac implantable electronic devices (CIED) implantation has been used increasingly in the last decades. These devices are known to increase survival and improve the life quality of patients suffering from cardiac rhythm diseases. As would be expected, the increase in number of implanted devices is also associated with an increase in the procedure related complications such as infection, malfunction, dislocations and vessel obstruction with a subsequent need to treat such complications without affecting long term prognosis of the patients. However, such procedural complications may affect the life expectancy of the patients due to the increase in morbidity and mortality [1, 2]. Various studies show infection rates of CEIDs between 1% and 7%. Likewise, the rate of dysfunctional leads reaches 7% [3–7]. The gold standard for treating CIED infection is device explanation as there is no therapy successful enough to eradicate the infection [8, 9]. There are many tactics to extract dysfunctional or infected leads. For example, simple traction or continuous traction or using polymer or metal sheath systems with or without locking stylets. Recently, many new devices were familiarized using laser or radiofrequency. However, these methods and devices are both expensive and time-consuming and require highly trained and skilled operators. Using laser-assisted percutaneous extraction of leads is correspondingly associated with serious life-threatening complications. We report our experience in CEID extraction using the 80 Hz high frequency laser sheaths to evaluate its safety and efficacy in octogenarian patients and its use should be limited to a younger group of patients or not due to the periprocedural consequence.

## Methods

This is a retrospective study of octogenarian patients who underwent laser-assisted lead extraction (LLE) (GlideLight laser sheath, Philips, San Diego, USA) between September 2013 and January 2020. The study protocol was approved by the institutional research ethics board at Heidelberg University, Germany. Consecutive 270 patients were included. Patients were divided into two groups. Octogenarian group and non-octogenarian group. The Data was gathered from patients treated between September 2013 and January 2020 and is retrospectively analyzed. Patients were referred either from external hospitals or from our electrophysiological outpatient clinic. Lead extraction was indicated if one of the following complications was diagnosed: Pocket infection, Device-related endocarditis, Pain, abandoned or non-functioning leads and blood vessel obstruction. The presence of redness with or without purulent discharge from the device pocket or device erosion was considered as device infection especially if associated with pain. CIED-related endocarditis was defined as the presence of bacteremia or sepsis in the absence of another recognizable source of infection or the presence of bacterial vegetations on the leads or the heart valves. If the patient is complaining of pain related to CEID or its Leads, the device and its leads may be explanted or translocated to alleviate the symptoms. Also, the presence of symptomatic obstruction of blood vessels may be treated with lead extraction of the non-functional leads or if the leads are not in use (abandoned leads). In case of new leads implantation, abandoned or non-functioning leads would be extracted to decrease the burden of the total lead number. During the procedure, the leads are examined and put under simple traction as a primary method for removal. If simple traction is not successful, the operator would use a laser sheath to extract the lead. As a standard measure, we perform all lead extractions in the operating theater and under general anesthesia continuous invasive monitoring of blood pressure and vital signs. After opening the device pocket, the device is extracted and the leads cut which allows the insertion of a suitable lead locking stylet into the inner coil lumen. After that a suture is tied around the insulation of the lead and the locking stylet. The laser

sheath (Glide Light 80 Hz, 14 or 16 French) is then advanced over the lead locking stylet until it emerges from the other side of the laser sheath. Under fluoroscopic guidance, the Laser sheath is then gradually advanced under traction over the lead while applying laser to the surrounding tissues until the lead is completely freed. During the whole process, transesophageal echocardiography was used to monitor the extraction and to identify any possible complications. The procedure was considered successful, if all leads are removed. If the extracted devices are not infected and if there are no contraindications, a new device was implanted from the same side using the laser sheath as a rail. Patients who present with local or systemic infection will not receive a new device unless they are pace-maker dependent. In these cases, epicardial pacemaker is implanted thought an inferior pericardiotomy as described by Al-Maisary et al. [10] the patient is discharged and during his follow up, an electrophysiological re-evaluation of would take place and a more suitable device will be implanted after remission of the infection. We retrospectively analyzed our electronic database including all patients who underwent laser assisted lead extraction. Data involving patients´ characteristics with the indications for lead extraction was collected. Also, data collection included the following variables: age and type of extracted leads, details of procedure, intra- and postoperative complications, in-Hospital mortality up to 30 days and pre- and postoperative medications. The procedure was considered successful, if all targeted leads are removed. Meanwhile, the failure to remove all targeted leads was considered as partial success. If no lead could be removed, procedure failure was declared. The Patients were divided into two groups. The first group (OG) represents the octogenarian group and included patients who are 80 or more years old. The second group (YG)included the patients who are younger than 80 years old. Any differences between these two groups were investigated. Data analysis for Demographic Data, medical history, and lead-related characteristics were presented as frequency (%) for categorical variables and as mean plus or minus standard deviation for normally distributed continuous variables. For continuous variables t tests were used. For nominal variables $\chi 2$ test or Fisher exact were used. A p value equal to or less than 0.05 was considered significant.

## Results

From September 1, 2013 to January 31, 2020 a total of 270 patients underwent percutaneous laser-assisted lead extraction in our center. Patient characteristics can be seen in Table 1.

**Table 1. Demographics.**

| | Total | OG | YG | P Value |
|---|---|---|---|---|
| Patients | n = 270 | 38 | 232 | |
| Age | 65±15 | 83±3.0 | 61±14 | 0.0001 |
| Sex, % male(n) | 74.0(200) | 89.4(34) | 71.5(166) | 0.21 |
| Body mass index | 27.0± 5.3 | 26.5±3.6 | 27.0±5.5 | 0.59 |
| Diabetes mellitus, %(n) | 29.6(80) | 0.05(16) | 27.5(64) | 0.10 |
| EF, % | 35.8± 14.7 | 44.8±13.7 | 40.3±14.8 | 0.08 |
| NYHA functional class III | 71(26.2%) | 15(39.4%) | 67(28.8%) | 0.18 |
| Coronary artery disease | 112(41.1%) | 26(68.4%) | 112(48.2%) | 0.02 |
| Arterial hypertension | 217(80.3%) | 32(84.2%) | 185(68.5%) | <0.01 |
| Renal insufficiency | 47(17.4%) | 11(28.9%) | 36(15.5%) | 0.07 |
| ICD | 195(72.2%) | 23(60.5%) | 172(74.1%) | 0.08 |

ICD = implantable cardioverter-defibrillator; n = number; EF = ejection fraction.

**Table 2. Indications for lead extraction.**

| | Total | OG | YG | P Value |
|---|---|---|---|---|
| Pocket infection | 96(35.5%) | 25(65.7%) | 71(30.6%) | < 0.001 |
| Endocarditis | 49(18.1%) | 6(15.7%) | 43(18.5%) | 0.68 |
| Bacteremia or Sepsis | 26(10.7%) | 2(5.2%) | 24(10.3%) | 0.32 |
| Functional abandoned lead | 10(3.7%) | 1(2.6%) | 9(3.8%) | 0.70 |
| Non-functional lead | 113(41.8%) | 5(13.1%) | 108(46.5%) | <0.001 |
| Pain at device or insertion site | 2(0.7%) | 0 | 2(0.8%) | >0.05 |
| Venous stasis or occlusion | 4(2.4%) | 0 | 4(1.7%) | >0.05 |

The mean age of patients was 65±15 years. The mean age of the patients in the OG was 83 ±3.0, meanwhile, the mean age of the YG was 61±14 with a significant difference in comparison to the OG. Of these patients, 200 were male (74.0%) with no statistical difference between both groups. The body mass index was 27.0± 5.3 and 29.6% had diabetes mellitus. The ejection fraction was 35.8±14.5% and only 71 (26.2%) patients had NYHA class III. Coronary heart disease was seen in 112(41.1%) of patients and 80.3% of patients had arterial hypertension. However, the OG showed a significant difference in the parameter compared to the YG. Only 17.4% presented with renal insufficiency (glomerular filtration rate (GFR) below 90 ml/min). Most of the patients (72.2%) had an ICD. The indications for lead extraction in this study are presented in Table 2, Figs 1 and 2.

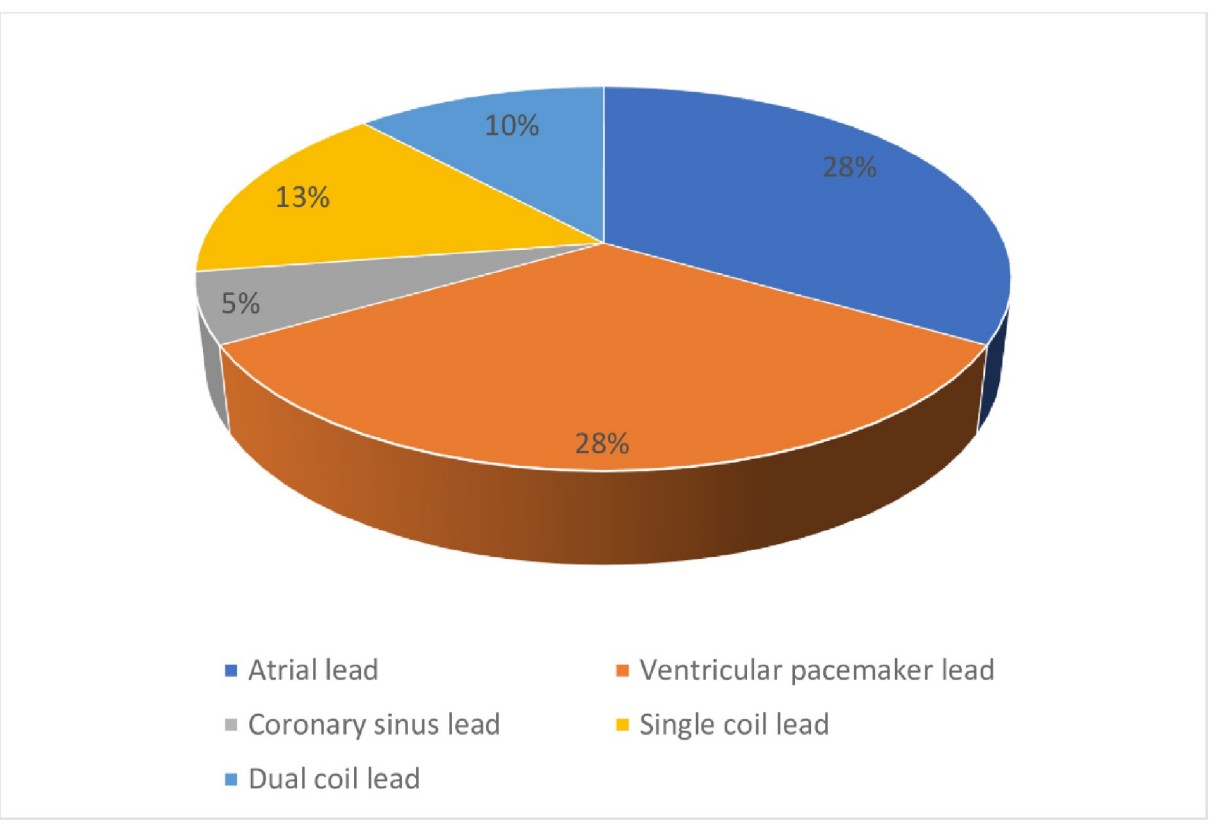

**Fig 1. Lead demographics in OG.**

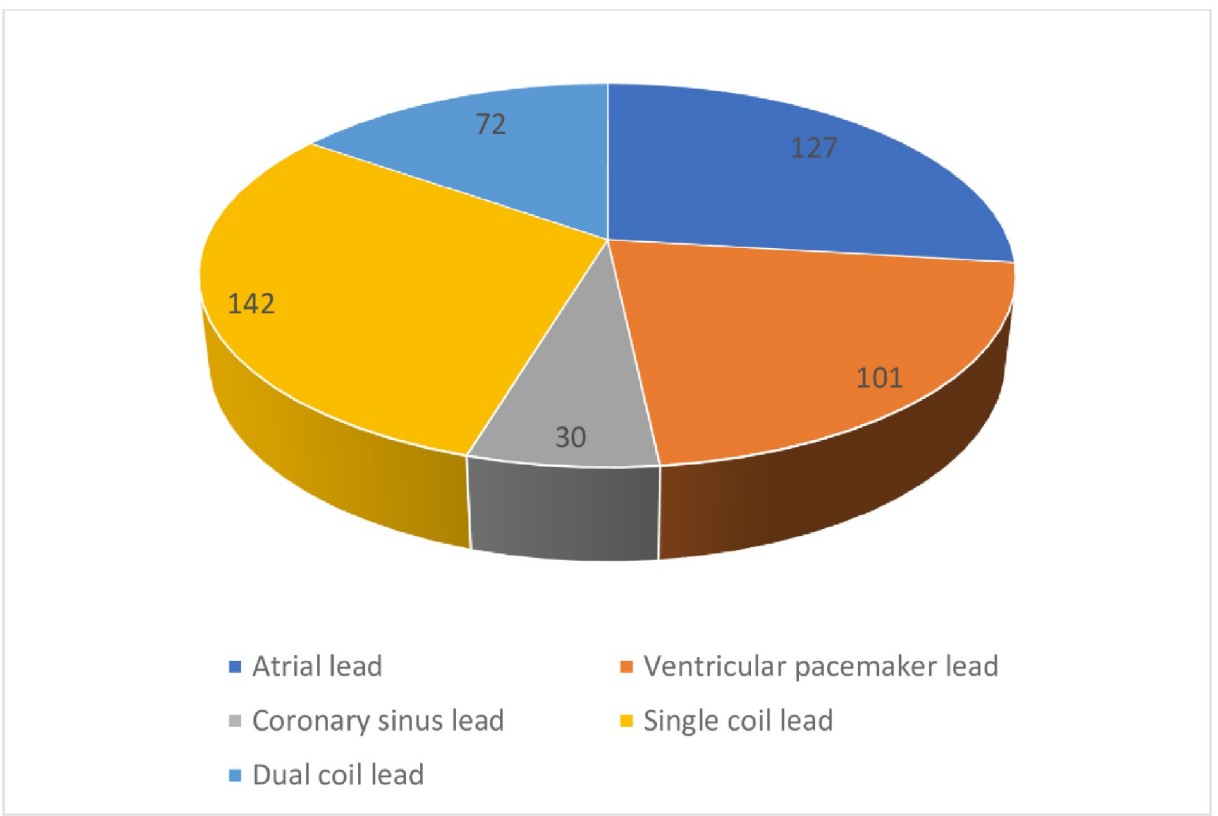

**Fig 2. Lead demographics in YG.**

The Most common indication for extraction was Nonfunctional lead (41.8%, n = 113) followed by pocket infection with (35.5%, n = 96) and endocarditis (18.1%, n = 49). However, pocket infection and nonfunction leads were seen more in the YG. Sepsis or bacteremia related to lead or device implantation were present in 10.7% of patients. Functional abandoned leads were an indication for lead extraction in 3.7% of patients. Only 2.4% of patients had venous stasis or occlusion and 2 patients asked for device removal due to chronic pain.

A total number of 556 leads underwent laser extraction with a mean time from implantation of 103.34± 65.70 months with significant difference between both groups (Table 3). Single coil ventricular leads and atrial leads were the most frequently encountered leads (27.87% both) followed by dual coil leads. There were significantly more left ventricular pacing leads in the OG but the single coil leads were significantly more in the YG. Coronary Sinus leads were the least frequently removed leads (6.3%).

As shown in Table 4, most of the Leads (92.6%) were extracted successfully. For patients, the percentage of complete successful removal of all leads was 90.0%, for partial success 7.8% (n = 21) and only 2.2%(n = 6) of the patients had no leads removed (Fig 3). Only one patient in the YG suffered from a superior vena cava tear requiring open surgery. In this patient, two leads were extracted. One lead was a right atrial lead and the other was a right ventricular pacing lead. Both leads were 60-month-old. After lead extraction. A new pleural effusion was detected and the patient became unstable. After sternotomy, a superior vena cava tear was discovered. One patient died due to pulmonary embolism. This patient had a vegetation on the tricuspid valve which embolized the lungs. No intra-operative mortality was recorded. A total

**Table 3. Lead characteristics.**

| | Total | OG | YG | P Value |
|---|---|---|---|---|
| Number of treated leads (Mean±Standard deviation) | 556 | 84(2.21±0.77) | 472(2.03±0.97) | 0.292 |
| Mean time from implantation, in months (Mean±Standard deviation) | 103.34± 65.70 | 112.4±80.8 | 101.8±62.9 | 0.365 |
| Lead type, n (Mean±Standard deviation within the group) | | | | |
| Atria lead (Mean±Standard deviation) | 155(0.57±0.56) | 28(0.73±0.44) | 127(0.54±0.57) | 0.055 |
| Ventricular pacemaker lead (Mean±Standard deviation) | 129(0.47±0.63) | 28(0.73±0.68) | 101(0.43±0.62) | 0.006 |
| Coronary sinus lead (Mean±Standard deviation) | 35(0.13±0.35) | 5(0.13±0.34) | 30(0.12±0.36) | 0.971 |
| Single coil lead (Mean±Standard deviation) | 155(0.57±0.56) | 13(0.34±0.48) | 142(0.61)±0.69 | 0.021 |
| Dual coil lead (Mean±Standard deviation) | 82(0.26±0.44) | 10(0.26±0.44) | 72(0.31±0.51) | 0.595 |

of 77 leads were extracted (2.02 lead per patient) in the OG and 435 leads (1.87 lead per patient) in the YG (Fig 4). Most of the extracted leads in the OG were equally right atrial und right ventricular leads with 28 leads for each. Only 13 single coil leads and 10 dual coil leads were removed in the octogenarian group and only 5 left ventricular leads were removed in the octogenarian group. In the YG, 142 single coil leads were removed which counted for the mostly removed leads followed by 127 right atrial leads and the right ventricular leads with 101 leads. Also, the number of removed dual coil leads was 72 leads and only 30 coronary sinus leads were removed.

## Discussion

In this study, 264 procedures were successfully performed in both study groups. The main difference between these two groups was only age which did not restrict the use of laser sheaths for lead extraction. Also, the variety of patient characteristics, the indications for device extraction did not appear to restrict the use of laser sheath use for lead extraction, success rate, or safety in both groups. This indicates that laser assisted lead extraction was safe in elderly as well as in young patients. As reported by Gould et al. and Deckx et al., the main reported risk factors for adverse events during laser assisted lead extraction procedures are infection, low body mass index, female sex, and long implantation duration but it did not evaluate the risk of higher age as a risk factor [11, 12].

The in-hospital 30-day deaths due to laser assisted lead extraction were 0% in both YG and OG, which are superior to the previous reports of 0.3% for in-hospital death [13]. Also, no intraoperative deaths were recorded in both groups, which is better than the recorded deaths of 0.28% [13]. Only one patient in the YG suffered a superior vena cava tear which was repaired after immediate thoracotomy. The rate of essential arterial hypertension and coronary artery disease is higher in the OG group but such a finding is expected in this group but it did

**Table 4. Outcome of laser lead extraction.**

| | Total | OG | YG | P Value |
|---|---|---|---|---|
| Outcome per lead | | | | |
| Success, n (%) | 512(92.08) | 77(91.6) | 435(92.1) | 0.88 |
| Failure, n (%) | 44(7.9) | 7(8.3) | 37(7.8) | |
| Outcome per patient | | | | |
| Complete Success, n (%) | 243(90.0) | 33(86.8) | 210(90.5) | 0.48 |
| Partial success, n (%) | 21(7.8) | 4(10.5) | 17(7.3) | |
| Failure, n (%) | 6(2.2) | 1(2.6) | 5(2.1) | |

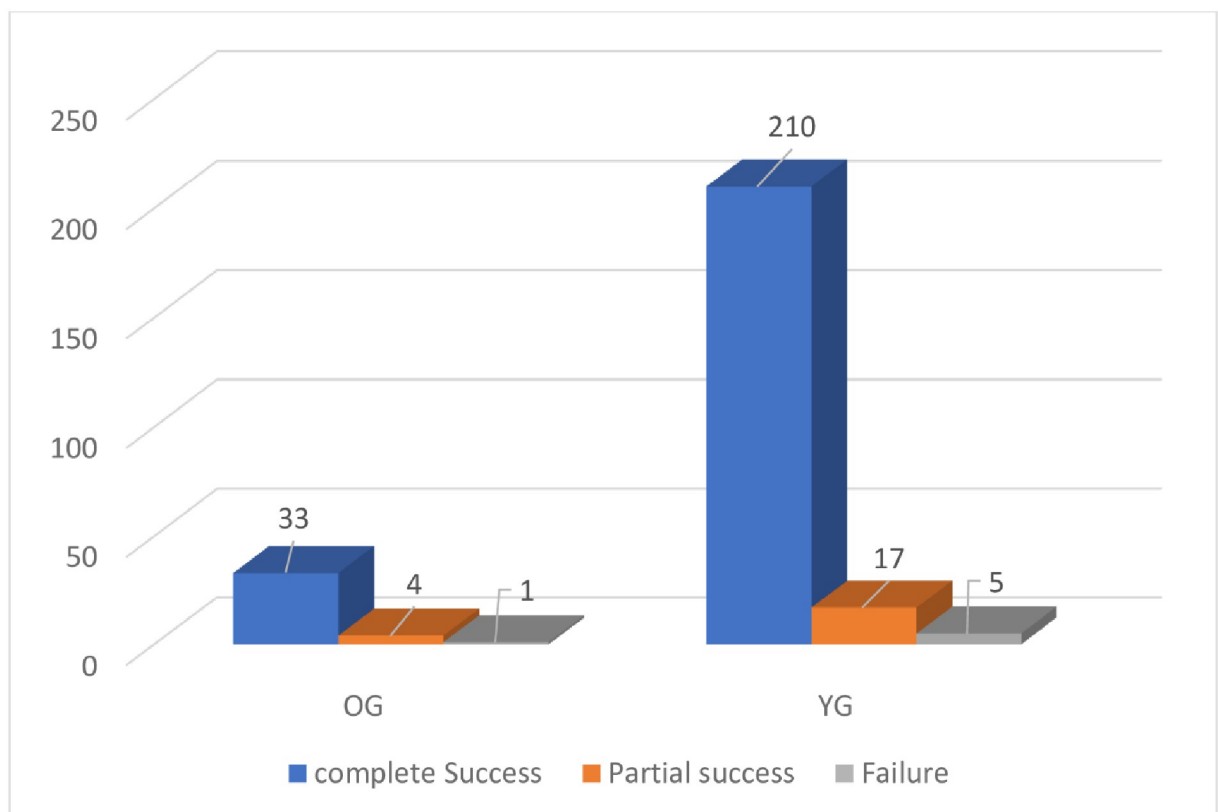

**Fig 3. Outcome of laser extraction per patient.**

not affect the outcome of the procedure [14]. Among the indications for lead extraction, pocket infection was more significantly seen in OG compared to the YG group. This may be due to the more tissue friability in the elderly which causes pocket perforation with subsequent pocket infection. Other possible causes may include the more difficult body hygiene or the weaker immune system, but these are theories that need further studies to prove. On the other hand, the number of extracted non-functional leads was significantly more in YG. These findings can be referred to the fact that the operator prefers to leave a sterile non-functional lead in the elderly non-touched to avoid the complications, while on the other hand, a more aggressive approach is used in the younger patient to lower the burden of non-functional leads. However, Rijal et al. analyzed the difference between these two strategies (Extracting versus abandoning sterile non-functional or recalled leads) and they found no difference in outcomes [15]. In the case of pain at device or insertion site or Venous stasis or occlusion, we did not have enough data to support a significant difference. Lead demographics showed no difference between these two groups. Even the age of the leads showed no difference between the groups. Man can say that the era of very old leads, which accumulated before the introduction of lead laser extraction, has ended [16]. The mean age of leads is 103.34± 65.70 months which means that these leads were implanted mostly after the introduction of laser assisted lead extraction in 1998 [17] and the introduction of second-generation laser sheath (SLS II) in 2002. The data shows a success in lead extraction of 92.08% and a success per patient of 90.0% which is close to the international success rate per lead of 96.5% and clinical success per patient of 97.7% [13]. In both parameters, no difference was noted between the study groups.

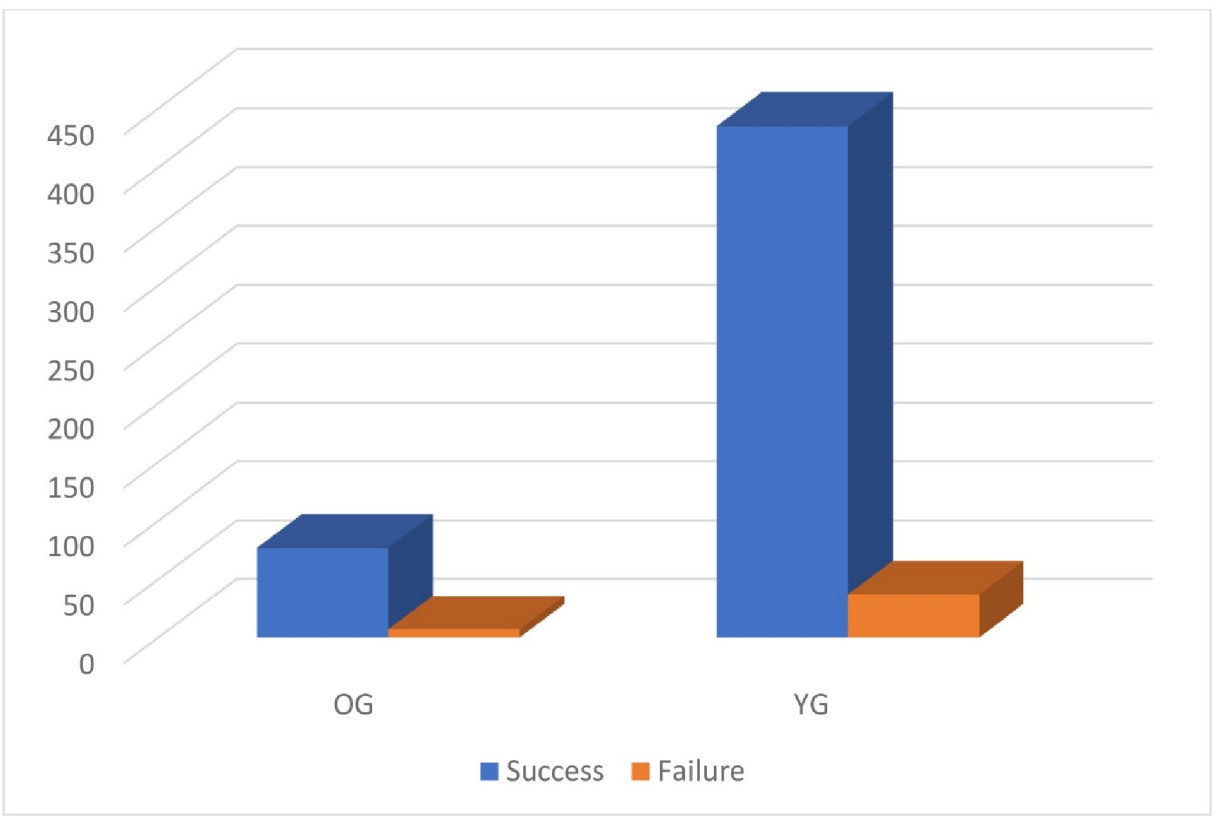

**Fig 4. Outcome of laser extraction per lead.**

Many investigators also studied the outcomes after transvenous lead extraction in octogenarians. Kutarski et al., Jacheć et al. and Giannotti et al. [18–20] studied the safety of lead extraction in octogenarians using mechanical tools and no laser was used. They concluded that lead extraction in octogenarians is also safe as in younger patients. On the other hand, Burger et al. [21] investigated the safety of powered extraction sheaths in octogenarians. In their study, additional mechanical rotational sheaths or femoral snares were used to achieve the success which is similar to our finding in which we used only laser sheaths.

## Conclusion

Transvenous laser-assisted lead extraction of cardiac implantable electrical devices in elderly patients is a safe and effective procedure. In the over 80s, the incidence of major complications in comparison with younger patients was equal and with at least a similar efficacy of the procedure and no procedural-related deaths. Age should not be a limit to withhold laser assisted lead extraction in the elderly patients.

## Study limitations

This is a single center study with a limited number of cases that might not reflect the real-world scenarios. Also, the setting and the experience could differ from one center to another. The mortality and morbidity may be due to the long standing local perioperative care and standard performance of the procedure but may be also biased through the limited number of cases.

## Supporting information

**S1 Data.**
(XLSX)

## Author Contributions

**Conceptualization:** Sameer Al-Maisary, Raffaele De Simone.

**Data curation:** Sameer Al-Maisary, Gabriele Romano, Jamila Kremer.

**Formal analysis:** Sameer Al-Maisary.

**Investigation:** Sameer Al-Maisary, Jamila Kremer.

**Methodology:** Sameer Al-Maisary, Jamila Kremer.

**Supervision:** Matthias Karck, Raffaele De Simone, Rawa Arif.

**Validation:** Matthias Karck.

**Visualization:** Matthias Karck, Rawa Arif.

**Writing – original draft:** Sameer Al-Maisary.

**Writing – review & editing:** Sameer Al-Maisary.

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
