## [Decision Letter · Decision Letter 0]

21 Nov 2022

PONE-D-22-26350Octogenarian patients and laser-assisted lead extraction: should we put a limit?PLOS ONE

Dear Dr. Al-Maisary,

Thank you for submitting your manuscript to PLOS ONE. After careful consideration, we feel that it has merit but does not fully meet PLOS ONE’s publication criteria as it currently stands. Therefore, we invite you to submit a revised version of the manuscript that addresses the points raised during the review process.

We look forward to receiving your revised manuscript.

Kind regards,

Redoy Ranjan, MBBS, MRCSEd, Ch.M., MS (CV&TS), FACS

Academic Editor

PLOS ONE

Journal Requirements:

2. During our internal evaluation of the manuscript, we found significant text overlap between your submission and previous work in the introduction.

Please revise the manuscript to rephrase the duplicated text, cite your sources, and provide details as to how the current manuscript advances on previous work. Please note that further consideration is dependent on the submission of a manuscript that addresses these concerns about the overlap in text with published work.

We will carefully review your manuscript upon resubmission and further consideration of the manuscript is dependent on the text overlap being addressed in full. Please ensure that your revision is thorough as failure to address the concerns to our satisfaction may result in your submission not being considered further.

Reviewers' comments:

Reviewer's Responses to Questions

**Comments to the Author**

1. Is the manuscript technically sound, and do the data support the conclusions?

Reviewer #1: No

Reviewer #2: No

Reviewer #3: Yes

2. Has the statistical analysis been performed appropriately and rigorously? 

Reviewer #1: I Don't Know

Reviewer #2: No

Reviewer #3: I Don't Know

3. Have the authors made all data underlying the findings in their manuscript fully available?

Reviewer #1: Yes

Reviewer #2: Yes

Reviewer #3: Yes

4. Is the manuscript presented in an intelligible fashion and written in standard English?

Reviewer #1: Yes

Reviewer #2: Yes

Reviewer #3: Yes

5. Review Comments to the Author

**Reviewer #1:** The study is a single center experience in transvenous lead extraction. The message the authors are trying to give is that age should not be the barrier. This message has been seen in other contemporary publications (see below).The study is retrospective. We are not told if there was patients who had an indication for lead extraction and were not done for whatever reason. Retrospective studies have inherent bias to what was essentially successful.

The problem with this paper is the small number of patients who are older than 80 years.

The authors have not done an appropriate literature search before writing their manuscript. Their most recent reference was dated 2019.

I strongly advice the authors look at similar publications on the topic with significantly larger numbers.

Some of the recent published manuscripts on this subject are below. After reading these the authors need to see if they can add to the literature anything else that has not been observed in these publications

Transvenous lead extraction: The influence of age on patient outcomes in the PROMET study cohort.

Akhtar Z, Elbatran AI, Starck CT, Gonzalez E, Al-Razzo O, Mazzone P, Delnoy PP, Breitenstein A, Steffel J, Eulert-Grehn J, Lanmüller P, Melillo F, Marzi A, Leung LWM, Domenichini G, Sohal M, Gallagher MM.

Pacing Clin Electrophysiol. 2021 Sep;44(9):1540-1548. doi: 10.1111/pace.14310. Epub 2021 Aug 5.

PMID: 34235772

Transvenous lead extraction in patients with systemic cardiac device-related infection-Procedural outcome and risk prediction: A GALLERY subgroup-analysis.

Chung DU, Burger H, Kaiser L, Osswald B, Bärsch V, Nägele H, Knaut M, Reichenspurner H, Gessler N, Willems S, Butter C, Pecha S, Hakmi S; GALLERY investigators.

Heart Rhythm. 2022 Oct 11:S1547-5271(22)02489-4. doi: 10.1016/j.hrthm.2022.10.004. Online ahead of print.

PMID: 36240993

Safe and effective transvenous lead extraction for elderly patients utilizing non-laser and laser tools: a single-center experience in Japan.

Okada A, Tabata H, Shoda M, Shoin W, Kobayashi H, Okano T, Yoshie K, Kato K, Saigusa T, Ebisawa S, Motoki H, Kuwahara K.

Heart Vessels. 2021 Jun;36(6):882-889. doi: 10.1007/s00380-020-01761-3. Epub 2021 Jan 4.

PMID: 33394103

Indications for transvenous lead extraction and its procedural and early outcomes in elderly patients: a single-center experience.

Ząbek A, Boczar K, Dębski M, Pfitzner R, Ulman M, Holcman K, Kostkiewicz M, Musiał R, Lelakowski J, Małecka B.

Pol Arch Intern Med. 2020 Mar 27;130(3):216-224. doi: 10.20452/pamw.15182. Epub 2020 Feb 10.

PMID: 32041927

Risk Factors and Long-Term Survival of Octogenarians and Nonagenarians Undergoing Transvenous Lead Extraction Procedures.

Jacheć W, Polewczyk A, Segreti L, Bongiorni MG, Kutarski A.

Gerontology. 2021;67(1):36-48. doi: 10.1159/000511358. Epub 2020 Nov 26.

PMID: 33242867

**Reviewer #2:** I read your article with great interest. The study was a retrospective study of 270 patients who underwent lead extraction between 2013-2020. The research question is reasonable; however, it is not uncommon for the patients with the age between 80-90 to undergo lead extraction procedure with the reasonable indications and thorough benefit-risk discussion with the patients and their family.

1. It would be an interesting descriptive study if the authors describe more in detail regarding the reasons of “partial success or failure” in the OG group. This is important because it reflects the bias of treatment leading to the favorable outcome. For example, the procedure was aborted in the OG group after stucking at the SVC level in extracting. This surely would make the procedure safe for the patients and there was no mortality in the OG group. In such a case, that OG patient did not truly undergo extraction to its entirety. It therefore does not support the conclusion the authors made that it is safe for octogenarians to undergo this high-risk procedure. If so, it should be mentioned in the discussion part. For CS lead and dual-coiled ICD, they pose significant risks if the lead has been dwelling in the body for more than 10 years, but it might not have a significant risk or be too difficult to extract if the duration of the implantation was less than that.

2. Table 1 should provide more detail in comorbidities e.g. prior PCI, CABG (might lower the risk of extraction in some patients due to the lower risk of cardiac tamponade esp if the tear occurred at the SVC level where pericardium still covers).

3. In the discussion part, the limitation of the study should be discussed more in detail as this is a singer-centered experience which may not be generalizable to other centers with different setup/experience in extraction procedure.

4. The statement: Only one patient suffered from SVC tier [the authors might misspell it; should be “tear”??]. Please describe in detail about this case; OG or YG group, dual-coil ICD lead or other types of the lead, how old the lead was, etc.

5. One patient died due to pulmonary embolism; this one is in the OG or YG group? Please describe in detail how/when it happened in the postoperative period and what was a likely etiology of this mortality.

**Reviewer #3:** Major issues:

1-Tables can be arranged in a simpler form. Inclusion of graphs and charts would be more effective in providing a visual comparison between the groups. This is specifically in reference to tables 3 and 4.

2- How do you explain the difference in death rates between your study and 0.3% in previous studies?

3-Where does your study account for risk factors of complications mentioned in discussion? Risk factors mentioned included body mass index, female sex. Were they accounted for during the selection process for the study? Were pts with lower mass index, female and having longer ICD implants excluded from selection for a laser extraction?

Minor issues:

Grammatical Errors

6. PLOS authors have the option to publish the peer review history of their article (what does this mean?). If published, this will include your full peer review and any attached files.

Reviewer #1: No

Reviewer #2: No

Reviewer #3: No

---

## [Author Response · Author response to Decision Letter 0]

21 Dec 2022

Dear Redoy Ranjan,

We thank the editors for giving us the opportunity to review our manuscript. As recommended by the editors, we performed a professional English language editing of the paper and we hope that we reach the journals standard. The Introduction section was already rewritten to avoid any overlap with own previous works. If you see any further text reproduction, please indicate it to me. Also, the previous similarities in the introduction section were with another paper talking a completely different issue in the field of CIED. So, the topic is different and because of that I can not provide advancement to the previous work. Regarding the data availability, I uploaded a data table for data availability. 

We thank the reviewers for the constructive comments on the manuscript. We will detail in our response below how we plan to address the reviewer comments.

Reviewer#1

Comment #1:

We are not told if there were patients who had an indication for lead extraction and were not done for whatever reason. Retrospective studies have inherent bias to what was essentially successful.

Answer to comment #1:

Thank you for your Comment. We as a reference center do not refuse to do operation even in high risk patient as long as we have a correct indication and a patient approval.

Comment #2:

Retrospective studies have inherent bias to what was essentially successful.

Answer to comment #2:

I agree with you.

Comment #3:

The problem with this paper is the small number of patients who are older than 80 years.

Answer to comment #3:

The number of patients over 80 is small But we were glad to have such a number.

Comment #4:

The authors have not done an appropriate literature search before writing their manuscript. Their most recent reference was dated 2019. I strongly advice the authors look at similar publications on the topic with significantly larger numbers. Some of the recent published manuscripts on this subject are below. After reading these the authors need to see if they can add to the literature anything else that has not been observed in these publications.

Answer to comment #4:

We thank you for your concerns. Our comments to the publication are noted below:

Transvenous lead extraction: The influence of age on patient outcomes in the PROMET study cohort.

Akhtar Z, Elbatran AI, Starck CT, Gonzalez E, Al-Razzo O, Mazzone P, Delnoy PP, Breitenstein A, Steffel J, Eulert-Grehn J, Lanmüller P, Melillo F, Marzi A, Leung LWM, Domenichini G, Sohal M, Gallagher MM.

Pacing Clin Electrophysiol. 2021 Sep;44(9):1540-1548. doi: 10.1111/pace.14310. Epub 2021 Aug 5.

PMID: 34235772

This also retrospective study showed the 30-day mortality was most significant in the octogenarian and least in the young patients. However, our data showed similar mortality in both groups. 

Transvenous lead extraction in patients with systemic cardiac device-related infection-Procedural outcome and risk prediction: A GALLERY subgroup-analysis.

Chung DU, Burger H, Kaiser L, Osswald B, Bärsch V, Nägele H, Knaut M, Reichenspurner H, Gessler N, Willems S, Butter C, Pecha S, Hakmi S; GALLERY investigators.

This retrospective subgroup study did not analyses octogenarian patients specifically. It analeysd patient with infected CIED as a risk factor.

Safe and effective transvenous lead extraction for elderly patients utilizing non-laser and laser tools: a single-center experience in Japan.

Okada A, Tabata H, Shoda M, Shoin W, Kobayashi H, Okano T, Yoshie K, Kato K, Saigusa T, Ebisawa S, Motoki H, Kuwahara K.

This retrospective study with a smaller patients’ groups as our showed promising results that support our finding. However, the study involved both laser and mechanical sheaths and as mentioned the groups were smaller. 

Indications for transvenous lead extraction and its procedural and early outcomes in elderly patients: a single-center experience.

Ząbek A, Boczar K, Dębski M, Pfitzner R, Ulman M, Holcman K, Kostkiewicz M, Musiał R, Lelakowski J, Małecka B.

Pol Arch Intern Med. 2020 Mar 27;130(3):216-224. doi: 10.20452/pamw.15182. Epub 2020 Feb 10.

PMID: 32041927

This retrospective study comparing two groups of young and octogenarian patients showed a with significantly higher non-procedure related 30 day mortality. On the countrary we proved that our patients showed no difference

Risk Factors and Long-Term Survival of Octogenarians and Nonagenarians Undergoing Transvenous Lead Extraction Procedures.

Jacheć W, Polewczyk A, Segreti L, Bongiorni MG, Kutarski A.

Gerontology. 2021;67(1):36-48. doi: 10.1159/000511358. Epub 2020 Nov 26.

PMID: 33242867

In this retrospective study, , leads were more often removed using standard extraction techniques: simple traction and mechanical dilatators in octogenarians and laser use in octogenarians was not the focus of the study.

Reviewer#2

Thank you for reviewing our article and for your positive feedback. We will respond to each point respectively below.

Comment #1:

It would be an interesting descriptive study if the authors describe more in detail regarding the reasons of “partial success or failure” in the OG group. This is important because it reflects the bias of treatment leading to the favorable outcome. For example, the procedure was aborted in the OG group after stucking at the SVC level in extracting. This surely would make the procedure safe for the patients and there was no mortality in the OG group. In such a case, that OG patient did not truly undergo extraction to its entirety. It therefore does not support the conclusion the authors made that it is safe for octogenarians to undergo this high-risk procedure. If so, it should be mentioned in the discussion part. For CS lead and dual-coiled ICD, they pose significant risks if the lead has been dwelling in the body for more than 10 years, but it might not have a significant risk or be too difficult to extract if the duration of the implantation was less than that.

Answer to comment #1:

Lead laser extraction is a challenging procedure and the complexity of lead extraction varies for patient to another. We cannot, from statistical view, involve such variations in a study which is supposed to be based on clear facts. The partial success per patient was defined as the inability to remove all leads. In the case of the octogenarian with failure to extract the leads, the laser sheath was advanced over the RV-lead down to the right ventricle. The cause for failure was that we could not free the tip of the lead and so we could not remove the lead. This means, that the laser sheath passed all the susceptible areas in the venous system without causing any complication. Because of that, we thought it is fair enough to include the patient in the study. About the comment on the CS and dual coil lead, we agree we your sightful comment.

Comment #2:

2. Table 1 should provide more detail in comorbidities e.g. prior PCI, CABG (might lower the risk of extraction in some patients due to the lower risk of cardiac tamponade esp if the tear occurred at the SVC level where pericardium still covers).

Answer to comment #2:

Thank you for the comment. Adding new variable to the study as PCI and CABG means we have to go through the entire data again to collect these variables which is very time-consuming and we did not believe it could have an influence on the results. We believe that having a previous PCI does not protect a against laser complications. On the other hand, a previous CABG could reduce the risk of laser. However, a perforation to the right pleura cavity through the subclavian vein or the superior vena cava could fatal. 

Comment #3:

In the discussion part, the limitation of the study should be discussed more in detail as this is a singer-centered experience which may not be generalizable to other centers with different setup/experience in extraction procedure.

Answer to comment #3:

Thank you for the advice. We added a new section titled study limitations based on your comment.

Comment #4

The statement: Only one patient suffered from SVC tier [the authors might misspell it; should be “tear”??]. Please describe in detail about this case; OG or YG group, dual-coil ICD lead or other types of the lead, how old the lead was, etc.

Answer to comment #4:

Thank you for the comment. We corrected the spelling mistake. In this patient, two leads were extracted. One lead was a right atrial lead and the other was a right ventricular pacing lead. Both leads were 60 month old. After lead extraction. A new pleural effusion was detected and the patient became unstable. After sternotomy, a superior vena cava tear was found. We added this to the manuscript.

Commnet #5:

One patient died due to pulmonary embolism; this one is in the OG or YG group? Please describe in detail how/when it happened in the postoperative period and what was a likely etiology of this mortality.

Answer to comment #5:

Thank you for the comment. This patient had a vegetation on the tricuspid valve which embolized to the lungs. We added this to the manuscript. The patient became unstable 2 hours after lead extraction and resuscitated in the intensive care and then token to the operation theater where he underwent open thrombectomy followed by ECMO implantation but he eventually died.

Reviewer #3:

Thank you for reviewing our article. Below you will and the answer to your kind comments

Comment #1:

Tables can be arranged in a simpler form. Inclusion of graphs and charts would be more effective in providing a visual comparison between the groups. This is specifically in reference to tables 3 and 4.

Answer to comment #1:

Thank you for the advice. We added charts to tables 3 and 4 as you suggested.

Comment #2

How do you explain the difference in death rates between your study and 0.3% in previous studies?

Answer to comment # 2:

This may be due to the long-standing experience in this field and the standard that we put and the good trained stuff to early detect any complications. Also, this is a small series so that it might represent the whole death rate overall.

Comment #3:

 Where does your study account for risk factors of complications mentioned in discussion? Risk factors mentioned included body mass index, female sex. Were they accounted for during the selection process for the study? Were pts with lower mass index, female and having longer ICD implants excluded from selection for a laser extraction?

Answer to comment #3:

In. this study we didn’t make any selection. All the patient who underwent laser lead extraction were included. After that, we analyzed the patients data to find the risk factor. So no one was excluded.

We hope the we answered all the questions properly.

With best regards.

Sameer Al-Maisary

---

## [Decision Letter · Decision Letter 1]

7 Mar 2023

PONE-D-22-26350R1

Octogenarian patients and laser-assisted lead extraction: should we put a limit?

PLOS ONE

Dear Dr. Al-Maisary,

Thank you for submitting your manuscript to PLOS ONE. After careful consideration, we feel that it has merit but does not fully meet PLOS ONE’s publication criteria as it currently stands. Therefore, we invite you to submit a revised version of the manuscript that addresses the points raised during the review process.

ACADEMIC EDITOR:

The authors are thanked for this submission to PLOS ONE. After a critical external peer review by three experts and considering the overall reviewers' comments, I reinforce improving your paper's clarity and presentation of the discussion section based on recent literature with a large sample and acknowledgement of concerns raised by reviewers. PLOS ONE's publication criteria considered methodological rigour and ethical standards, regardless of the paper's novelty.

Please see the attached reviewer comments detail below.

We look forward to receiving your revised manuscript.

Kind regards,

Redoy Ranjan, MBBS, MRCSEd, Ch.M., MS (CV&TS), FACS

Academic Editor

PLOS ONE

Journal Requirements:

Reviewers' comments:

Reviewer's Responses to Questions

**Comments to the Author**

1. If the authors have adequately addressed your comments raised in a previous round of review and you feel that this manuscript is now acceptable for publication, you may indicate that here to bypass the “Comments to the Author” section, enter your conflict of interest statement in the “Confidential to Editor” section, and submit your "Accept" recommendation.

Reviewer #4: All comments have been addressed

Reviewer #5: All comments have been addressed

Reviewer #6: (No Response)

2. Is the manuscript technically sound, and do the data support the conclusions?

Reviewer #4: Yes

Reviewer #5: Yes

Reviewer #6: Partly

3. Has the statistical analysis been performed appropriately and rigorously? 

Reviewer #4: Yes

Reviewer #5: N/A

Reviewer #6: I Don't Know

4. Have the authors made all data underlying the findings in their manuscript fully available?

Reviewer #4: Yes

Reviewer #5: Yes

Reviewer #6: No

5. Is the manuscript presented in an intelligible fashion and written in standard English?

Reviewer #4: Yes

Reviewer #5: Yes

Reviewer #6: No

6. Review Comments to the Author

**Reviewer #4:** The topic of the study is important because, according to the common opinion, TLE in patients > 80 years old may be more dangerous and some patients are disqualified for TLE only because of their age. There are many publications on this topic, many of which use more conventional lead extraction techniques, but with similar results. The study is based on the small number of patients >80y (38 patients), however, a certain advantage of it is a uniform (laser) extraction technique and performing procedures in organizational conditions ensuring maximum safety. In everyday practice, the stepwise approach and cross-over strategy (from simple traction, utility polypropylene sheaths of different sizes, and in the absence of progress of lead dilatation – up to utility mechanical rotational threaded tip sheaths or laser sheaths) are more often used. The authors' results confirm previous reports, enriching our knowledge that also extraction of leads using the laser technique is not more dangerous in octogenarians than in younger patients.

TLE appears to be easier in the elderly. Other authors explain this by a weaker reaction of the endothelium of veins and heart structures to the presence of leads, less intensive development of connective tissue scar around the lead and less intense processes of its mineralization and calcification.

It seems that if the available material is modest, the discussion should be based on publications containing a much larger number of 80- and 90-year-olds.

I suggest that you pay attention to such reports as:

Jacheć W, Polewczyk A, Segreti L, Bongiorni MG, Kutarski A. Risk Factors and Long-Term Survival of Octogenarians and Nonagenarians Undergoing Transvenous Lead Extraction Procedures. Gerontology. 2021;67:36-48. (80-90y -549 patients and over 90 years of age -35 patients).

Giannotti Santoro M, Segreti L, Zucchelli G, Barletta V, Fiorentini F, Di Cori A, De Lucia R, Bongiorni MG. Transvenous lead extraction: Efficacy and safety of the procedure in octogenarian patients. Pacing Clin Electrophysiol. 2020;43:382-387 (202 patients >80)

Kutarski A, Polewczyk A, Boczar K, Ząbek A, Polewczyk M. Safety and effectiveness of transvenous lead extraction in elderly patients. Cardiol J. 2014;21:47-52. (192 patients over 80y)

Burger H, Hakmi S, Petersen J, Yildirim Y, Choi YH, Willems S, Reichenspurner H, Ziegelhoeffer T, Pecha S. Safety and efficacy of transvenous lead extraction in octogenarians using powered extraction sheaths. Pacing Clin Electrophysiol. 2021;44:601-606 (71 patients over 80y)

**Reviewer #5:** Lead extraction procedures have increased in recent years due to the augmented number of cardiac implantable electronic devices, a term which includes implantable cardioverter-defibrillators (ICDs)

and permanent pacemakers (PMs). The main reasons for lead extraction are infection and malfunction of the device and/or catheters either after a first implant or a substitution/upgrade procedure.

The paper is a interesting paper. Is ok the remember that the indication for the lead extraction in not relatel to the age of the patients. Is possible to accept with the corrections.

**Reviewer #6:** In this manuscript authors report a retrospective comparison of two population (one of octogenarian patients, one of patients < 80 years old) undergoing TLE for various indications. Study aims are potentially interesting as the target demographics for this kind of procedure has progressively aged in recent decades. However, this manuscript holds several significant flaws in study ideation, conduction and reporting. English language needs major revision before consideration. Study population, especially for the target demographics (patients < 80 years old) is small, significantly reducing result generalizability.

Evidence regarding this topic is broadly available in papers with higher standards and larger populations:

- https://www.sciencedirect.com/science/article/pii/S0914508719302953

- https://pubmed.ncbi.nlm.nih.gov/23799557/

- https://pubmed.ncbi.nlm.nih.gov/33394103/

- https://www.ahajournals.org/doi/full/10.1161/CIRCEP.111.964270

- https://www.karger.com/Article/Abstract/511358

Minor:

ABSTRACT:

- Methods: “Consecutive 270 patients” should say “270 consecutive patients”

- Results: apart from mortality, briefly, other complications should be cited

- Conclusion: “In octogenarian” should say “In octogenerians”

MAIN:

- Methods:

- “Consecutive 270 patients” should say “270 consecutive patients”

- “The of redness” should be rephrased

- Statistical analysis should include all statistical tests used

- Please define “renal insufficiency”

- Results:

- “Lead demographics” table should be renamed to “Lead characteristics”, are numbers relative to dwelling time in months? There is no description or legend.

- Figure 1-2-3: please add numerical values, graphs are not useful as presented.

7. PLOS authors have the option to publish the peer review history of their article (what does this mean?). If published, this will include your full peer review and any attached files.

Reviewer #4: **Yes: **Andrzej Kutarski

Reviewer #5: No

Reviewer #6: No

---

## [Author Response · Author response to Decision Letter 1]

18 Mar 2023

Reviewer#4

Comment #1:

The topic of the study is important because, according to the common opinion, TLE in patients > 80 years old may be more dangerous and some patients are disqualified for TLE only because of their age. There are many publications on this topic, many of which use more conventional lead extraction techniques, but with similar results. The study is based on the small number of patients >80y (38 patients), however, a certain advantage of it is a uniform (laser) extraction technique and performing procedures in organizational conditions ensuring maximum safety. In everyday practice, the stepwise approach and cross-over strategy (from simple traction, utility polypropylene sheaths of different sizes, and in the absence of progress of lead dilatation – up to utility mechanical rotational threaded tip sheaths or laser sheaths) are more often used. The authors' results confirm previous reports, enriching our knowledge that also extraction of leads using the laser technique is not more dangerous in octogenarians than in younger patients.

Answer to comment #1:

Thank you for this comment. Adding to the existing knowledge is our goal and we try to optimize our techniques.

Comment #2:

TLE appears to be easier in the elderly. Other authors explain this by a weaker reaction of the endothelium of veins and heart structures to the presence of leads, less intensive development of connective tissue scar around the lead and less intense processes of its mineralization and calcification.

Answer to comment #2:

I agree with you. 

Comment #3:

It seems that if the available material is modest, the discussion should be based on publications containing a much larger number of 80- and 90-year-olds.

I suggest that you pay attention to such reports as:

Jacheć W, Polewczyk A, Segreti L, Bongiorni MG, Kutarski A. Risk Factors and Long-Term Survival of Octogenarians and Nonagenarians Undergoing Transvenous Lead Extraction Procedures. Gerontology. 2021;67:36-48. (80-90y -549 patients and over 90 years of age -35 patients).

Giannotti Santoro M, Segreti L, Zucchelli G, Barletta V, Fiorentini F, Di Cori A, De Lucia R, Bongiorni MG. Transvenous lead extraction: Efficacy and safety of the procedure in octogenarian patients. Pacing Clin Electrophysiol. 2020;43:382-387 (202 patients >80)

Kutarski A, Polewczyk A, Boczar K, Ząbek A, Polewczyk M. Safety and effectiveness of transvenous lead extraction in elderly patients. Cardiol J. 2014;21:47-52. (192 patients over 80y)

Burger H, Hakmi S, Petersen J, Yildirim Y, Choi YH, Willems S, Reichenspurner H, Ziegelhoeffer T, Pecha S. Safety and efficacy of transvenous lead extraction in octogenarians using powered extraction sheaths. Pacing Clin Electrophysiol. 2021;44:601-606 (71 patients over 80y)

Answer to comment #3:

Thank you for suggestions. We added these publications and discussed it in the discussion as you proposed.

Reviewer#5:

Comment:

Lead extraction procedures have increased in recent years due to the augmented number of cardiac implantable electronic devices, a term which includes implantable cardioverter-defibrillators (ICDs)

and permanent pacemakers (PMs). The main reasons for lead extraction are infection and malfunction of the device and/or catheters either after a first implant or a substitution/upgrade procedure.

The paper is a interesting paper. Is ok the remember that the indication for the lead extraction in not relatel to the age of the patients. Is possible to accept with the corrections.

Answer to comment:

Thank you for reviewing our article. We did our best to correct the pitfalls to reach a high standard publication. 

Reviewer #6:

Comment #1:

In this manuscript authors report a retrospective comparison of two population (one of octogenarian patients, one of patients < 80 years old) undergoing TLE for various indications. Study aims are potentially interesting as the target demographics for this kind of procedure has progressively aged in recent decades. However, this manuscript holds several significant flaws in study ideation, conduction and reporting. English language needs major revision before consideration.

Answer to comment #1:

We thank you for your concerns. We did a detail revision of the paper and corrected the grammatic mistakes as you advised. 

Comment #2:

Study population, especially for the target demographics (patients < 80 years old) is small, significantly reducing result generalizability.

Answer to comment #2:

We agree we you that the study population is small, but this is because the number of octogenarians undergoing laser extraction is not huge and the participation of every experience will enrich our knowledge to better treat these patients. 

Comment #3:

Evidence regarding this topic is broadly available in papers with higher standards and larger populations:

- https://www.sciencedirect.com/science/article/pii/S0914508719302953

- https://pubmed.ncbi.nlm.nih.gov/23799557/

- https://pubmed.ncbi.nlm.nih.gov/33394103/

- https://www.ahajournals.org/doi/full/10.1161/CIRCEP.111.964270

- https://www.karger.com/Article/Abstract/511358

Answer to comment #3:

Thank you for mentioning these papers. We will discuss each one below:

- https://www.sciencedirect.com/science/article/pii/S0914508719302953

Atsuhiko et al discussed the safety of laser lead extraction in Japanese population with a low BMI and a smaller collective in comparison to our study which has more collective and normal to slightly elevated BMI.

- https://pubmed.ncbi.nlm.nih.gov/23799557/

Kutarski et al paper was included in our discussion in its reviewed version. It has showed good result after mechanical extraction. No laser was used.

- https://pubmed.ncbi.nlm.nih.gov/33394103/

This retrospective study with a smaller patients’ groups as our showed promising results that support our finding. However, the study involved both laser and mechanical sheaths and as mentioned the groups were smaller.

- https://www.ahajournals.org/doi/full/10.1161/CIRCEP.111.964270

This study from 2011 also investigated the safety of TLE in octogenarian. The authors reported 100% success in all patient but did not mention how many leads were left. They mentioned the removed leads. I think at that time, Laser extraction was a new tool and over the years, our experience has advanced and our approach also. So this paper is relatively good for that time but did not give us the perspectives that we need now. 

- https://www.karger.com/Article/Abstract/511358

Jacheć et al paper is included n the revised version of the paper. It investigated mechanical lead extraction.

Comment #4:

Methods: “Consecutive 270 patients” should say “270 consecutive patients”

Answer to comment #4:

Thank you for your correction. We change the sentence as you adviced.

Comment #5:

Results: apart from mortality, briefly, other complications should be cited.

Answer to comment #5:

Thank you for the comment. We added the other complications to the abstract.

Comment #6:

Conclusion: “In octogenarian” should say “In octogenerians”

Answer to comment #6:

Thank you for the comment. We checked the spelling and it was correct so we did not change it.

Comment #7:

The of redness” should be rephrased

Answer to comment #7:

Thank you for the correction. We rephrased the sentence. 

Comment #8:

Statistical analysis should include all statistical tests used

Answer to comment #8:

Thank you for the comment. We included the used statistical tests to the methods. 

Comment #9:

Please define “renal insufficiency”

Answer to comment #9:

Thank you for the comment. We added a definition to the results section.

Comment #10:

“Lead demographics” table should be renamed to “Lead characteristics”, are numbers relative to dwelling time in months? There is no description or legend.

Answer to comment #10:

Thank you for the comment. We changed lead demographics to lead characteristics. The dwelling time is in months. we only added (in) to increase the clarity.

Comment #11:

Figure 1-2-3: please add numerical values, graphs are not useful as presented.

Answer to comment #10:

Thank you for the comment. We added numerical values to the graphs.

We hope the we answered all the questions properly.

With best regards.

Sameer Al-Maisary

---

## [Decision Letter · Decision Letter 2]

10 Apr 2023

Octogenarian patients and laser-assisted lead extraction: should we put a limit?

PONE-D-22-26350R2

Dear Dr. Al-Maisary,

We’re pleased to inform you that your manuscript has been judged scientifically suitable for publication and will be formally accepted for publication once it meets all outstanding technical requirements.

Kind regards,

Redoy Ranjan, MBBS, MRCSEd, Ch.M., MS (CV&TS), FACS

Academic Editor

PLOS ONE

**Additional Editor Comments (optional):** The authors are thanked for this submission to PLOS ONE. After a critical external peer review by the experts and considering the overall reviewers' comments and authors' responses, your manuscript meets PLOS ONE's publication criteria, i.e. fulfils the methodological rigour and ethical standards.

Review Comments to the Author

Reviewer #4: The manuscript has been revised and supplemented. All reviewer’s suggestions were completed. I have no more critical comments.

Reviewer #5: Dear Authors, the paper is OK but is necessary to note that the safety of the lead extraction procedures un the elderly patients has been evaluated also with others techniques. Transvenous lead extraction: The influence of age on patient outcomes in the PROMET study cohort.

Akhtar Z, Elbatran AI, Starck CT, Gonzalez E, Al-Razzo O, Mazzone P, Delnoy PP, Breitenstein A, Steffel J, Eulert-Grehn J, Lanmüller P, Melillo F, Marzi A, Leung LWM, Domenichini G, Sohal M, Gallagher MM.

Pacing Clin Electrophysiol. 2021 Sep;44(9):1540-1548

---

## [Editor Report · Acceptance letter]

13 Apr 2023

PONE-D-22-26350R2 

Octogenarian patients and laser-assisted lead extraction: should we put a limit? 

Dear Dr. Al-Maisary:

I'm pleased to inform you that your manuscript has been deemed suitable for publication in PLOS ONE. Congratulations! Your manuscript is now with our production department. 

Kind regards, 

on behalf of

Dr. Redoy Ranjan 

Academic Editor

PLOS ONE